# QTL Mapping and a Transcriptome Integrative Analysis Uncover the Candidate Genes That Control the Cold Tolerance of Maize Introgression Lines at the Seedling Stage

**DOI:** 10.3390/ijms24032629

**Published:** 2023-01-30

**Authors:** Ru-yu He, Tao Yang, Jun-jun Zheng, Ze-yang Pan, Yu Chen, Yang Zhou, Xiao-feng Li, Ying-zheng Li, Muhammad-Zafar Iqbal, Chun-yan Yang, Jian-mei He, Ting-zhao Rong, Qi-lin Tang

**Affiliations:** Maize Research Institute, Sichuan Agricultural University, Chengdu 611130, China

**Keywords:** maize introgression line, low-temperature, genetic map, RNA-seq, candidate genes, functional prediction

## Abstract

Chilling injury owing to low temperatures severely affects the growth and development of maize (*Zea mays*.L) seedlings during the early and late spring seasons. The existing maize germplasm is deficient in the resources required to improve maize’s ability to tolerate cold injury. Therefore, it is crucial to introduce and identify excellent gene/QTLs that confer cold tolerance to maize for sustainable crop production. Wild relatives of maize, such as *Z*. *perennis* and *Tripsacum dactyloides,* are strongly tolerant to cold and can be used to improve the cold tolerance of maize. In a previous study, a genetic bridge among maize that utilized *Z. perennis* and *T. dactyloides* was created and used to obtain a highly cold-tolerant maize introgression line (MIL)-IB030 by backcross breeding. In this study, two candidate genes that control relative electrical conductivity were located on MIL-IB030 by forward genetics combined with a weighted gene co-expression network analysis. The results of the phenotypic, genotypic, gene expression, and functional verification suggest that two candidate genes positively regulate cold tolerance in MIL-IB030 and could be used to improve the cold tolerance of cultivated maize. This study provides a workable route to introduce and mine excellent genes/QTLs to improve the cold tolerance of maize and also lays a theoretical and practical foundation to improve cultivated maize against low-temperature stress.

## 1. Introduction

Chilling injury is a major abiotic stress that affects crop growth, development, and yield. As an important crop, maize (*Zea mays* L.) is sensitive to cold stress, particularly at the seedling stage [1,2,3,4]. Maize planted in high-latitude temperate regions, such as the United States maize belt, Central Canada, Central and Northern Europe, the Atlantic Coast, Central Asia, Russia, the Indian subcontinent, and Northern China, often suffer from low-temperature and cold damage before and after the seedlings emerge, which results in a large area of yield reduction during the later period [5]. Understanding the genetic mechanisms and introducing/identifying cold-tolerant genes/QTLs is a prerequisite to improve maize against chilling damage at the seedling stage and cultivate cold-tolerant maize cultivars for sustainable production.

Maize is very sensitive to low temperatures during the transition from heterotrophic growth to autotrophic growth [6]. At this stage, when maize seedlings are subjected to low temperatures, there is a decrease in the height and leaves of the maize seedlings. Huang et al. conducted a low-temperature treatment on this stage of maize seedlings in an artificial climate control chamber and showed that low temperatures had a substantial influence on the relative water content, relative leaf number, relative RL, relative seedling length, relative total fresh weight, and relative PH. The symptoms included browning and wilting of the mesocotyl and coleoptyl sheaths, water-stained leaves, and the stunted growth of seedlings [7]. Strigens et al. [8] used a field natural environment and an indoor artificial climate incubator to treat maize seedlings with low temperatures, and the plants grew relatively well. The photosynthetic efficiency, chlorophyll content, SDW, and RDW were also significantly affected. Cold stress also changes the cell homeostasis and increases the levels of reactive oxygen species, which generally results in the oxidation of membrane lipids and damage to cell membranes. The biofilm is the interface between cells and the external environment. Many life activities and physiological functions are conducted on the biofilm, and all types of adverse effects on cells primarily originate from the membrane. The membrane system is the first part of the plant to be damaged under low-temperature stress [9]. SOD, CAT, and POD are cytoprotective enzymes whose changes within cells reflect the potential of varying plant groups/species under study to tolerate cold [10]. In addition, CBF is one of the most important TFs since it can interact with the DRE/CRT *cis*-acting elements to regulate the expression of various stress-induced genes. Previous studies that explored the molecular mechanisms of the cold stress response in maize were primarily based on *Arabidopsis thaliana*. The proteins encoded by *COR* and the other genes are strongly hydrophilic and stable, which could involve the resistance to damage owing to cell dehydration caused by low-temperature stress. Overexpression of the *CBF* gene in *A. thaliana* [11,12] or the transfer of *CBF* homologous genes into other plants can enhance the resistance of recipient plants to low-temperature stress. Song et al. showed that the *ADC* gene upregulated by cold (*Citrus sinensis ADC; CsADC*) elevated the endogenous content of Put in sweet orange (*C. sinensis*), and the overexpression of *CsCBF1* led to the notable elevation of the expression of *CsADC* and levels of Put in sweet orange transgenic plants, along with a remarkable enhancement in cold tolerance compared with that of the WT [13]. Similarly, in rice (*Oryza sativa* L.), the intracellular accumulation of ROS caused by cold stress activates the TF ICE1 homolog protein *OsbHLH002*, which binds to the downstream gene *OsTPP1* and promotes its expression; this increased the content of cold-resistant metabolite trehalose in rice, which enabled the plants to resist cold stress [14]. Besides the activation of TFs/proteins, lipids are also involved in the tolerance to abiotic stress in plants [15,16]. Lipids, such as wax, cutin, and suberin, directly contribute to the alleviation of drought [17] by reducing cellular dehydration and lipid metabolism [15]. The overexpression of wax biosynthetic genes increases tolerance to drought [18,19,20], and plants that have been depleted of wax are less tolerant to drought [21,22,23]. In addition, phospholipase D alpha 1 (PLDα1) is an enzyme that hydrolyzes phospholipids and participates in multiple regulatory roles in the stress responses of plants. Takáč et al. concluded a quantitative proteomic analysis of two T-DNA insertion PLDα1 mutants of *A. Thaliana*, and the genetic depletion of PLDα1 was shown to also affect the proteins involved in the cell wall architecture, redox homeostasis, and ABA signaling. PLDα1 appears to be a protein that integrates cytosolic and plastidic protein translations, plastid protein degradation, and protein import into the chloroplast to regulate chloroplast biogenesis in *A. thaliana* [24].

Linkage mapping is a useful method to dissect the genetic basis of quantitative traits [25]. Currently, the mapping of low-temperature tolerance genes in maize seedlings primarily focuses on changes in the photosynthetic system, the morphology of shoots and roots, and biomass [8,26,27,28,29,30]. GWAS enable the detection of QTLs with a high-resolution approach [31] and have been used extensively to excavate the candidate genes that control targeted traits [32]. The use of GWAS in temperate maize inbred lines identified 36 QTLs under cold conditions [33]. Unterseer et al. [34] identified the signatures of selections that were specific to temperate dent and flint pools and identified genes under selective pressure that differed between the dent and flint pools by performing GWAS. The complex mapping populations of maize, such as connected populations, have also been used to map QTLs for cold tolerance [35]. A total of 650 families were used to map 43 QTLs, followed by the selection of three QTLs by a meta-analysis. Some studies identified the QTLs associated with cold tolerance at the germination stage [5,36,37] or during seedling emergence [38]. Yi et al. [39] performed a GWAS analysis on 406 recombinant inbred lines from a MAGIC population and found 858 SNP sites that were significantly associated with cold tolerance phenotypes. Most were related to chlorophyll and the maximum photosynthetic activity of photosystem II (Fv/Fm). Goering et al. [40] used an IBM Syn4 population and located two major QTLs that affected the chlorophyll concentration, leaf color, and tissue damage in maize seedlings that were treated at 4 °C in the dark for 8 h and then recovered at 24 °C for 24 h. Yi et al. [41] conducted a GWAS of 836 maize inbreds at the germination and seedling stages and found 187 significant SNPs that were integrated into 159 QTLs for emergence, and the traits were related to early growth. In recent years, the combined analysis of forward genetics and a WGCNA has become an effective method to decode the genes that control genetic information and the candidate genes that are associated with target traits. Guo et al. [42] used the strategies of GWAS and WGCNA to identify seven hub candidate genes involved in the seed RL of maize seedlings under drought stress. This strategy provided a reference to map the cold tolerance genes in MIL-IB030 seedlings.

Although significant progress has been made in the study of genes/QTLs related to cold tolerance in maize, there is little information on the mechanism of response of maize to cold stress and the expression of a regulatory network of maize cold tolerance genes. Additionally, the associated phenotypic traits are complex and controlled by multiple genes under different environments. Wild relatives of maize, such as *Z. perennis* and *Tripsacum dactyloides*, have many beneficial genes that are essential for resistance to cold, salt, diseases, and insect pests that are absent in maize [43,44]. Creating genetic bridges between crop and wild species is a sophisticated, direct way to overcome the barrier of hybridization and introduce exogenous genes into crops [45]. We previously constructed a genetic bridge (allohexaploid hybrid: MTP; 2n = 74 = 20M + 20P + 34T) among maize, *Z. perennis,* and *T. dactyloides*, which not only polymerized the genomes of the three species, but also identified excellent resistance genes of the three species that could be used to obtain cold-tolerant MILs by backcrossing [46,47]. In this study, the MILs were identified and screened at the seedling stage at 2 °C. Subsequently, 289 F_2:3_ populations were constructed using cold-tolerant MIL-IB030 and cold-sensitive MIL-IB021. In addition, RNA-Seq was performed using MIL-IB030 and the recurrent parent B73 after the plants were treated with low-temperature stress (2 °C) at the seedling stage. Forward genetics combined with a WGCNA strategy was used to study the genetic mechanism of cold tolerance in MIL-IB030 at the seedling stage, and the major QTLs and candidate genes that are related to cold tolerance traits in MIL-IB030 were located. This study lays the foundation for using MTP to introduce beneficial genes into maize, clone new cold-tolerant genes, and facilitate the breeding of cold-tolerant maize.

## 2. Results

### 2.1. Evaluation and Screening of the Tolerance of MILs to Cold

The abilities of 21 MILs to tolerate cold were evaluated by subjecting them to low-temperature stress (2 °C) compared with a control (25 °C) at the seedling stage (Appendix A). The phenotypic traits showed a skewed distribution with highly significant differences and strong correlations among the MILs (Appendix A). The 21 MILs were classified into cold tolerance grades using the membership function method, which resulted in one highly cold-resistant line, eight cold-resistant lines, seven cold-sensitive lines, and five highly cold-sensitive lines (Appendix A). Consequently, one highly cold-tolerant line (MIL-IB030) and one highly cold-sensitive line (MIL-IB021) were selected as parents to construct the 289 F_2:3_ populations, which located the major QTLs/genes by controlling low-temperature-stress-related traits at the seedling stage.

MIL-IB030 and MIL-IB021 and wild-type B73 were subjected to low-temperature stress at the seedling stage (2 °C). MIL-IB021 and B73 had stunted growth, with higher leaf wilting rates, water loss, and necrosis compared with those of MIL-IB030 (Figure 1A,C–J), which reflects the higher cold tolerance of MIL-IB030 compared to that of the wild type B73 and MIL-IB021. Other phenotypic and physiological indicators, including PH, RFW, SDW, REC, RL/PH, and RDW, performed significantly better than those of MIL-IB021 and B73 under normal and low-temperature-stress conditions (Appendix A).

### 2.2. Phenotypic Identification and Genetic Analysis of the F_2:3_ Populations

There were significant differences in the PH, REC, SFW, SDW, and RL/PH among the 289 F_2:3_ populations that were subjected to low-temperature stress (Table 1, Appendix A), and the coefficient of variation, kurtosis, and skewness for RL/PH were larger among the 289 F_2:3_ populations. A genetic analysis was performed to identify the phenotypic indices of the 289 F_2:3_ populations under cold stress, and all the phenotypic traits showed a normal distribution (Figure 2A). Therefore, these traits were quantitative traits that were controlled by multiple genes. The phenotypic indices of 289 F_2:3_ populations under low-temperature stress were analyzed by a PCA. It is worth noting that REC had the highest rate of contribution (Figure 2B), and follow-up studies should focus on the major QTLs of REC to identify candidate genes.

### 2.3. Construction of High-Density Genetic Linkage Maps and Mapping the QTLs That Controlled Seedling Stage-Related Traits in the F_2:3_ Populations

This study selected 2396 pairs of SSR primers that were evenly distributed in 10 linkage groups from the Maize GDB (http://www.maizegdb.org, V4 (accessed on 1 March 2019)). The results of parental polymorphism primer screening showed that there were 280 SSR markers that were highly polymorphic and had reliable results that were evenly distributed on the 10 chromosomes of maize (Appendix A). Consequently, a linkage genetic map was constructed of 289 F_2:3_ populations using QTL ICI Mapping 4.2 software. The full length of this linked genetic map was 7,449.51 cM, and the average genetic distance between the markers was 26.60 cM (Figure 3, Appendix A).

Subsequently, 25 QTLs that were closely linked to REC, PH, SFW, SDW, and RL/PH were detected in the F_2:3_ populations at low-temperature stress (Appendix A, Appendix A), and nine major QTLs were located that controlled seedling-related traits (Figure 4). *qTREC3-1*, *qTREC6-1,* and *qTREC8-1* were located on chromosomes 3, 6, and 8, respectively, and the LOD values were 3.08, 4.82, and 3.57, respectively, while the explainable rates of phenotypic contribution were 2.09, 3.46, and 6.03%, respectively. *qTSFW4-1* was located on chromosome 4 and had an LOD value of 6.51, and the explainable rate of phenotypic contribution was 8.77. *qTSDW2-1* and *qTSDW4-1* were located on chromosomes 2 and 4, respectively, and the LOD values were 3.72 and 3.99, respectively. The explainable phenotypic contribution rates were 1.0 and 1.41%, respectively. *qTRLRPH2-1* was located on chromosome 2 and had an LOD value of 7.41, and the explainable phenotypic contribution was 10.57% (Table 2).

### 2.4. RNA-Seq and Identification of the Candidate Genes

To explore the candidate genes responsible for cold tolerance in MIL-IB030, transcriptome sequencing was performed on MIL-IB030 and B73 after they had been subjected to cold stress (2 °C). The results showed that the base error rate of each sample was 0.02%, and the content of Q20, Q30, and GC was more than 98, 94, and 55%, respectively (Appendix A). There were 45 million bp of raw sequencing data, which was reduced to 43 million high-quality sequences by data filtering. The clean bases of each exceeded 6 G, indicating a high sequencing depth. More clean reads were mapped, and the reference genome was higher in 36 of the samples, among which the mapping of MIL-IB030 > 85% (Appendix A). Correlation and PCA analyses showed that there was high similarity in the patterns of expression between the samples within a group (Appendix A), and the differences between groups were significant. The samples within the groups clustered together, while different groups were dispersed.

Subsequently, the candidate genes that controlled REC, SDW, and SFW were located by forward genetics combined with WGCNA. This study identified 264 candidate genes that controlled the REC of MIL-IB030 under low-temperature stress through the map-based cloning of *qTREC3-1*, *qTREC6-1*, and *qTREC8-1*. A WGCNA was performed with the FPKM values of 264 candidate genes, and 264 genes could be divided into seven subgroups, which consisted of brown, green, yellow, red, blue, turquoise, and grey (Figure 5A). Green and yellow genes correlated the most highly with REC for subsequent analysis. Five hub genes were screened from the green module based on |KME|> 0.9 and TOM> 0.2 (Figure 5B), including *Zm00001d012313* (KME = 0.90, TOM = 0.39), *Zm00001d037538* (KME = −0.98, TOM = 0.33), *Zm00001d037590* (KME = 0.91, TOM = 0.29), *Zm00001d037605* (KME = 0.92, TOM = 0.26) and *Zm00001d011971* (KME = 0.96, TOM = 0.21) (Table 3). Four hub genes, including *Zm00001d012321* (KME = −0.97, TOM = 0.43) and *Zm00001d037551* (KME = 0.96, TOM = 0.35), *Zm00001d011879* (KME = −0.95, TOM = 0.31), and *Zm00001d037602* (KME = −0.92, TOM = 0.27), were screened from the yellow module (Figure 3C and Table 3).

This study used map-based cloning to identify 130 candidate genes that controlled the SDW (*qTSDW2-1* and *qTSDW4-1*). WGCNA was performed with the FPKM values of 130 candidate genes, and 130 genes could be divided into six subgroups, which were brown, green, yellow, blue, turquoise, and grey modules (Figure 5D). The yellow module genes correlated the most highly with SDW for subsequent analysis. The use of |KME|> 0.9 and TOM> 0.2 resulted in the screening of three hub genes from the yellow module (Figure 5E), including *Zm00001d052201* (KME = 0.96, TOM = 0.25), *Zm00001d052096* (KME = 0.94, TOM = 0.27), and *Zm00001d052097* (KME = 0.96, TOM = 0.25) (Table 3).

In addition, map-based cloning was used to identify 126 candidate genes that controlled SFW (*qTSFW4-1*). WGCNA was performed with the FPKM values of 126 candidate genes, and 126 genes could be divided into six subgroups, which were brown, green, yellow, blue, turquoise, and grey modules (Figure 5F). The yellow module genes correlated the most highly with SFW for subsequent analysis. Subsequently, two hub genes, *Zm00001d052201* (KME = 0.96, TOM = 0.25) and *Zm00001d052096* (KME = 0.94, TOM = 0.27), were screened from the yellow module (Figure 5G and Table 3). The functions of these hub genes were annotated using the COG, KOG, SwissProt, and NR databases, and the specific results are shown in Appendix A.

It is worth noting that REC had the highest rate of phenotypic contribution under low-temperature stress (Figure 2B). Thus, the candidate genes that controlled REC were the subject of focus. This study annotated *Zm00001d037590* and *Zm00001d012321* as protein disulfide isomerase 7 and nematode resistance protein-like HSPRO1, respectively. Additionally, the functional annotation of these genes was compared with their homologous genes in *A. thaliana*, Japanese rice*,* and rapeseed (*Brassica napus*) (Appendix A). The findings of this study suggest that *Zm00001d037590* and *Zm00001d012321* could be candidate genes that confer cold tolerance in MIL-IB030 at the seedling stage. Therefore, we explored the genetics and function of these two genes in detail.

### 2.5. Analyses of Gene Structure and Cis-Control Elements

The gene annotation files of the candidate genes were extracted, and the length and related information of the hub genes were tabulated. The *Zm00001d012321* gene was 1,425 bp long and contained one CDS region, and the *Zm00001d037590* gene was 4,643 bp long and included two UTR regions and CDS regions (Appendix A). The *Zm00001d012321* gene contains one exon and no intron, while the *Zm00001d037590* gene is composed of 12 exons and 11 introns. The statistics of the exon distribution of each hub gene are shown in Appendix A.

The extracted *cis*-acting elements in the upstream 2 kb sequence of the candidate genes are shown in Appendix A. The 2 kb upstream sequence of *Zm00001d012321* had one *cis*-acting element that contained MYB transcription factor binding and recognition sites, seven *cis*-acting elements related to the hormone response, five *cis*-acting elements relating to defense responses to stress, three *cis*-acting elements that were related to light responses, and one *cis*-acting element that was related to meristem expression. It was hypothesized that the candidate gene *Zm00001d012321* could be mediated by the regulation of hormonal signaling systems in MILs to regulate defenses against stress. The 2 kb upstream sequence of *Zm00001d037590* had 10 *cis*-acting elements that were related to hormone responses, seven *cis*-acting elements related to ABA responses, nine *cis*-acting elements related to stress defense responses, and *cis*-acting elements that are involved in the light response. These results suggested that the candidate gene *Zm00001d037590* was probably involved in the ABA-mediated hormone signal transduction system, which could confer cold tolerance in MIL-IB30. However, this merits additional verification.

### 2.6. Candidate Gene Cloning, Expression Pattern Analysis, and Candidate Genes That Were Homologously Cloned in MTP and Z. Perennis

*Zm00001d012321* in B73 and MIL-IB030 was amplified and sequenced to detect whether SNP mutations occurred in the candidate gene, and multiple base variations were identified in the *Zm00001d012321* gene region. These results led to changes in the amino acid sequence of this gene (Appendix A). *Zm00001d037590* in B73 and MIL-IB030 was amplified and sequenced, and there were multiple base deletion variants in the *Zm00001d037590* gene region. The eight base pairs starting from the 651st to 661st exon were deleted in the candidate gene of the mutant; bases 662 to 686 of exon 9 were deleted, and base 913 was mutated from G to A. These results led to changes in the amino acid sequence of the gene (Appendix A).

This study performed qRT-PCR to verify the patterns of expression of the candidate genes. The results showed that *Zm00001d012321* and *Zm00001d037590* in B73 exhibited a trend of increasing first and then decreasing when prolonging the time of treatment, while the level of expression of MIL-IB030 first decreased and then increased (Figure 5H). These preliminary findings indicate that the cold tolerance of MIL-IB030 was stronger than that of B73, which could be owing to the significantly higher level of expression of the candidate gene in MIL-IB030 than in B73.

Exploring the source of variation in the nucleotide sequences of candidate genes was crucial. Therefore, the homologous candidate genes were cloned from MTP and *Z. perennis* cDNA. Compared with the WT B73, *Zm00001d012321* showed multiple SNP variants in MTP, *Z. perennis*, and MIL-IB030, and these changes were non-synonymous mutations (Appendix A). *Zm00001d037590* had a 37 bp indel deletion in MTP, *Z. perennis*, and MIL-IB030, and this was also a non-synonymous mutation (Appendix A). The SNPs/indel mutation sites of *Zm00001d012321* and *Zm00001d037590* in MIL-IB030 were conclusively consistent with the gene sequence of the distant parent, *Z. perennis*. These results indicate that the source of variation in both candidate genes of MIL-IB030 is the stable inheritance of introgression from *Z. perennis*.

### 2.7. Subcellular Localization and Functional Verification of the Candidate Genes

This study also transiently expressed *Zm00001d037590* in the epidermal cells of tobacco (*Nicotiana benthamiana*) leaves to localize its protein within the cells. The results of confocal microscopy showed that the green fluorescence of *Zm00001d037590*-GFP fusion protein coincided with the red fluorescence of ER-mCherry (Figure 6A3) but did not coincide with the red fluorescence of NLS-mCherry (Figure 6A7). These results indicate that the *Zm00001d037590* protein is localized to the endoplasmic reticulum.

The WT B73, MIL-IB030, and EMS single-base mutant lines were subjected to low-temperature stress to validate the functions of candidate genes. The results showed that EMS4-13c465 (*Zm00001d037590*), EMS4-15d91a (*Zm00001d012321*), EMS4-0b1833 (*Zm00001d012321*), and EMS4-15d90b (*Zm00001d012321*) suffered severe damage at low temperatures, which manifested as leaf wilting, leaf browning, water loss, and cell death. The wild-type B73 also suffered some degree of low-temperature damage, but the degree of damage at low temperatures was not as severe as that of the EMS mutant. MIL-IB030 suffered the least from the low-temperature injury and grew normally at 2 °C. All the genotypes, including the WT B73, MIL-IB030, and the EMS mutant lines, grew normally at 25 °C (Figure 6B).

Furthermore, to confirm the reliability of the phenotypic results, this study compared the REC of the leaves of the different materials under study. The results showed that EMS4-13c465 (*Zm00001d037590*), EMS4-15d91a (*Zm00001d012321*), EMS4-0b1833 (*Zm00001d012321*), and EMS4-15d90b (*Zm00001d012321*) had the largest leaf REC, indicating that the leaf cell membranes were severely damaged under low-temperature stress. Damage to the leaf REC of WT B73 was intermediate, and the leaf REC indicated that MIL-IB030 was the least affected. These results indicate that MIL-IB030 had lower levels of cell membrane damage compared with the parent B73 and EMS mutant lines (Figure 6C–F), thereby revealing that MIL-IB030 exhibited improved cold tolerance.

The RT-PCR results showed that *Zm00001d037590* and *Zm00001d012321* had the highest levels of expression in MIL-IB030, followed by the WT B73 and the EMS mutant lines, respectively. Differences in relative gene expression among the three materials reached a significant level under low-temperature stress (Figure 6G–J). The phenotypic indicators, REC content of the leaves, and the profile of expression of the candidate genes led to the conclusion that *Zm00001d037590* and *Zm00001d012321* positively regulate cold tolerance in MIL-IB030.

### 2.8. Identification of the Gene Family among Different Varieties of Maize

In addition, this study compared the protein sequences of the candidate genes with those of their homologous genes that corresponded to the longest transcripts of maize inbred lines B97, CML52, CML247, DK105, Ki3, Ki11, Ky21, Mo17, Mo18, Ms71, NC350, Oh43, P39, W22, and *ZX-PI566673* and identified 31 paralogous genes of *Zm00001d012321* from these 15 maize inbred lines. Additionally, one paralogous gene was identified from *ZX-PI566673*, and two paralogous genes were identified from the remaining 14 inbred maize lines. The homologous genes of *Zm00001d012321* that were identified were highly similar in terms of gene length, gene structure, DNA, CDS, and protein sequences (Appendix A). Thereby, it is inferred that this gene is highly conserved in maize.

A total of 109 genes that were paralogous with *Zm00001d037590* were identified from the 15 maize inbred lines described above. Among them, seven paralogous genes were identified in maize inbred line B73. Eleven paralogous genes were identified in maize inbred line B97. Five paralogous genes were identified in maize inbred line DK105. Seven paralogous genes were identified in the maize inbred line Ki3. One paralogous gene was identified in the maize inbred line Ky21. Four paralogous genes were identified in the maize inbred line Mo18. Seven paralogous genes were identified in the inbred line Ms71 and the maize inbred lines NC350 and Oh43. Four paralogous genes were identified in the maize inbred line P39 and in *ZX-PI566673*. Eleven paralogous genes were identified in inbred line CML52. A total of 13 paralogous genes were identified in the maize inbred line Ki11. Eight paralogous genes were identified in maize inbred line CML247, and two paralogous genes were identified in the maize inbred line Mo17 (Appendix A).

A phylogenetic analysis was performed on the DNA sequences of the target gene *Zm00001d012321* and 31 HSP genes in the maize inbred lines (Figure 7A). The results of a phylogenetic analysis showed that the 31 HSP orthologous genes described above were divided into the two subfamilies HSP1 and HSP2. The genes from different maize inbred lines were highly similar or even identical in their gene length, gene structure, base sequence, CDS sequence, and protein sequence when they were in the same homology group. However, there were significant differences in these parameters among the different groups (HSP1 and HSP2). In addition, this study also showed that *Zm00001d012321* was closely related to the maize inbred lines oh43 and ky21.

A phylogenetic analysis of the protein sequences encoded by the target gene *Zm00001d037590* and 109 genes in the other maize inbreds (Figure 7C) showed that the 109 PDIL family members described above were divided into three subfamilies, namely PDIL1, PDIL2, and PDIL3. In addition, the members within a family showed similar gene structures, gene lengths, CDS sequences, and protein sequences, but the genes in different subfamilies clearly differed in these parameters. There were more members in the subgene family PDIL1, while the subgene family PDIL3 had the least members. Based on these results, it was hypothesized that the gene family members were differentiated to differing degrees during the evolutionary process.

In this study, the analysis of the selection pressure of the paralogous genes of *Zm00001d012321* and *Zm00001d037590* in the 16 maize inbred lines (Figure 7B,D) resulted in paralogs to *Zm00001d012321* and *Zm00001d037590*. Natural mutations in the homologous genes were significantly lower than those caused by non-natural environment/factors. The Ka/Ks values in each maize inbred line showed a trend of <1 and peaked at 0.5 and 0.7. This showed that *Zm00001d012321* and *Zm00001d037590* and their homologous genes in the inbred line of maize species were purified and selected and had been relatively conserved during natural evolution.

## 3. Discussion

### 3.1. Excavation of the Cold-Tolerant Resources of Wild Maize Materials Is an Important Strategy

Maize originated from Central and South America. It has poor tolerance to cold and is sensitive to cold damage. Directional genetic breeding of maize for high yields resulted in the loss of gene diversity and accumulation of deleterious alleles. The narrow genetic base of the existing germplasm hinders the further improvement of maize against many biotic and abiotic stresses, including cold stress. Transferring the cold-tolerant genes/QTLs from wild relatives into cultivated maize is an important strategy to broaden the genetic base of maize, expand the germplasm resources, and discover excellent stress-resistant genes for crop improvement. *Z. mays* subsp*. mexicana* has been domesticated for a long time and displays excellent resistance to cold, diseases and insects, salinity, and drought. In recent years, the use of wild resources for crop genetic improvement has been receiving increasing amounts of attention among breeders. Yang et al. [48] assembled high-quality maize and *Z. mays* subsp. *mexicana* genomes and observed that more than 10% of the maize genome showed introgressions from the *Z. mexicana* genome, indicating that *Z. mexicana* has contributed to the adaptation and improvement of maize. Recent research revealed that an introgressed fragment of teosinte [49] was involved in regulating maize phospholipid metabolism and flowering, suggesting that teosinte had a role in contributing to maize domestication and improving the abiotic stress tolerance of the maize crop. *Tripsacum dactyloides* (2n = 72) is a perennial forage species and possesses many excellent traits, including resistance to cold, diseases, insect pests, and salt. Hybrids of maize and *T. dactyloides* were more tolerant to cold compared with maize at the seedling stage, which indicates that *T. dactyloides* can improve the cold resistance of maize [50]. A large deletion variant fragment that confers abiotic stress resistance and is related to many agronomic traits was identified in the maize genome, and it was then verified as an introgression from *T. dactyloides* by the integrated approaches of cytogenetics, genomics, and population genetics [51]. These approaches revealed the contribution of *T. dactyloides* in the evolution and domestication of maize. In a previous study, the allohexaploid genetic bridge (MTP) among maize, *Z. perennis*, and *T. dactyloides* was created to ease gene flow and obtain a highly cold-tolerant MTP–maize introgression line by backcross breeding [46,47,52].

### 3.2. Major QTLs/Genes for Controlling Cold Tolerance in MILs

It is important to locate new genes to control cold tolerance in MILs. The construction of a genetic map is an important link in studying biological genomes and is the basis of gene mapping, gene cloning, and the identification of genome structures. In maize, SSR markers are considered to be the most suitable markers to construct genetic linkage maps and evaluate QTLs, owing to their high allelic variation and co-dominance. Ramekar et al. [53] used 907 analytical markers, including MUTD, SCARs, SSRs, and SNPs, to construct a genetic map of the maize RIL population. The results showed that the full length of the genetic map was 6248.2 cM; the average genetic distance between markers was 6.84 cM, and there were 24 QTLs related to grain yield and quality. In this study, the F_2:3_ molecular genetic linkage map covered 10 maize chromosomes with that were 7449.51 cM long in total and had an average genetic distance between markers of 26.60 cM. This map also met the requirements of high-density genetic linkage maps that were published internationally and were suitable for QTL mapping. Previous studies have shown that the major QTLs that control maize seedling-related traits under low-temperature stress are concentrated on chromosomes 1, 5, and 6 [54]. In recent years, GWAS have been used to study the genetic mechanism of cold tolerance in maize seedlings. Huang et al. [7] conducted GWAS on 10 cold tolerance traits of 125 maize inbred lines at the seedling and seed germination stages, and 43 SNPs were associated with cold tolerance. A total of 40 candidate genes related to cold tolerance were screened within 50 kb upstream and downstream of the 43 SNPs, but no SNPs that showed cold tolerance were screened at both the seedling and germination stages, which could be owing to the different genetic basis of cold tolerance at the seedling and seed germination stages. Hu et al. [5] used 243 RIL (B73xMol7) families to detect 15 QTLs, which were primarily distributed on chromosomes 4, 5, 6, 7, and 9, after treatment at 12 /16, 18/8, and 28 °C/24 h, respectively, with the relative germination rate and primary RL as indicators. Yi et al. [39] used GWAS analysis to construct 700 IBM populations using Mo17 and B73 and mined 858 SNPs sites that were significantly associated with the cold tolerance phenotype. Most of the QTLs were associated with chlorophyll and Fv/Fm. Yi et al. [41] conducted GWAS on 836 maize inbred lines after low-temperature stress at the germination and seedling stages, and 187 SNPs were detected, which were integrated into 159 QTLs. This study identified nine major QTLs that were distributed on chromosomes 2, 3, 4, 6, and 8, which controlled seedling-related traits at a low temperature of 2 °C (Figure 4). In addition, this study identified more major QTLs, which are more comprehensive than the previous research results. These results lay the foundation for the fine mapping of candidate genes.

In recent years, the combined analyses of forward genetics and WGCNA have become an effective method to decode the genes that control genetic markers and candidate genes associated with the target traits. Guo et al. [42] used this strategy with GWAS and WGCNA to identify seven candidate genes for the seedling RL of maize seedlings under drought stress. Ma et al. [55] used GWAS to identify nine candidate genes that were associated with salt tolerance in maize seedlings. Ma et al. [56] used forward genetics combined with a gene co-expression analysis to identify the gene *ZmHIPP* that controlled tolerance to lead in maize and analyzed its regulatory mechanism. In this study, forward genetics combined with WGCNA was used to excavate two candidate genes that controlled the leaf REC of MILs under low-temperature stress (Figure 5B,C and Table 3).

### 3.3. Exploring the Function of Candidate Genes Is Critical

It is crucial to explore the function of candidate genes. Goldberger et al. [57] were the first to utilize this by approach by isolating and purifying protein disulfide isomerase (*PDI*) from human liver tissue. Some *PDI* genes were then cloned from vertebrates and yeast and found to have various intracellular functions, such as contributing to cell stability, ion absorption, gene activation, and cell differentiation. Subsequently, various experiments confirmed that the *PDILs* could catalyze the oxidation, reduction, and isomerization of protein disulfide bonds, and it has molecular chaperone and anti-chaperone activities and can bind calcium and copper [58,59,60]. Houston et al. [61] showed that the diversity of *A. thaliana PDI*-like proteins was important to adapt to stressful environments. Environmental stress also induced a rapid increase in the expression of *PDI* in vegetative organs to defend against stress [62]. This study utilized subcellular localization experiments to show that *Zm00001d037590* is primarily expressed in the endoplasmic reticulum (Figure 6A). Studies have shown that *PDI* plays an important role in the abiotic stress response. Hatahet F et al. [63] reported that the *PDI* of rice seedlings was significantly upregulated after treatment with cadmium. Chen et al. [64] pretreated rice with glutathione before Hg^2+^ stress and found that the expression of *PDILs* in rice root cells was significantly upregulated, which confirmed that the *PDILs* were involved in the response of rice to Hg^2+^ stress. In addition, MTH1745 transgenic rice can improve the activity of antioxidant enzymes, effectively remove intracellular ROS, and alleviate the membrane lipid peroxidation damage caused by Hg^2+^. The content of NPT and GSH in transgenic plants was higher than that of the wild type [65]. The AtPDI1 protein encoded by the *At3g54960* gene of *A. thaliana* has been shown to have disulfide bond isomerase activity in vitro, and the *AtPDI1* deletion mutant (pdi) of *A. thaliana* can reduce the ability of plants to tolerate abiotic stress. *AtPDI1* enhances plant resistance by participating in systems that scatter reactive oxygen and transduce ABA signal transduction pathways [66]. This further demonstrates the role of PDI proteins in improving plant resistance. However, the molecular mechanism of the response of *PDI* in cold stress signals in maize is still unclear and merits additional study. This study has begun to deeply analyze the molecular regulatory mechanism and functional evolutionary characteristics of the bond candidate gene *ZmHSP2* by molecular genetic methods, and to develop molecular markers for marker-assisted selection breeding to expand the resource pool of germplasm for cold tolerance. In addition, the analytical results of the *cis*-elements in the promoter region of the candidate gene showed that 10 *cis*-acting elements were related to hormone responses. Seven *cis*-acting elements were related to the abscisic acid response; seven *cis*-acting elements were involved in the defense against stress, and nine *cis*-acting elements were related to the light response (Appendix A). It was inferred that the candidate gene *Zm00001d037590* could be involved in regulating the external stress response through the ABA signal transduction system network.

HSPs are involved in the plant responses to heat stress and extensively participate in other environmental stresses, such as stress, salinity, osmotic, cold, and oxidative stress [67]. HSPs function as molecular chaperones and play an important role in protecting plants from stress and restoring cellular homeostasis [68]. Many studies have confirmed that the primary HSPs can act as molecular chaperones, which explains why the biosynthesis of HSPs can confer the ability to withstand temperature stress. Low temperature is also associated with protein dysfunction and denaturing, which induce the accumulation of HSPs [69,70,71]. Many HSPs were found to respond to cold stress and were upregulated in *A. thaliana*, tobacco, maize, rapeseed, chicory (*Cichorium intybus*), poplar (*Populus* spp.), wheat (*Triticum aestivum*), and barley (*Hordeum vulgare*) [70,72,73]. *HSPs* are induced under situations of low-temperature stress and translocated into various cell organelles to protect them from cold stress [74]. Some *HSPs* specifically accumulated in tissues upon low-temperature exposure, as in poplar, where the *HSPs* accumulated in leaves [75]. In rice, low-temperature stress and a gradual decrease in the temperature from 15 to 0 °C at intervals of 5 °C upregulated *HSP95, HSP*75, and *HSP70*, which resulted in their accumulation in the chloroplast, since this is the part of plant that is most vulnerable to low temperatures [76]. Some of the *HSPs*, such as *HSP90* in wheat and *HSP60* and *HSP21* in sunflowers (*Helianthemum* spp.), are downregulated in response to cold stress [72,77]. A similar trend was also reported by Hlavackova et al. [70] and Rinalducci et al. [78], where the stability of Rubisco was associated with the downregulation of *HSP60* and *HSP*21 in winter wheat. In addition, the *HSP* gene that was expressed at low temperatures was cloned from tomato (*Solanum lycopersicum* L.), and its cDNA was a member of the *LMHSP* family; a sequence analysis of their CDS showed that they were highly homologous to the tomato genes *TOM66* and *TOM111*, which enhance cold tolerance in tomatoes [79]. Physiological changes caused by low-temperature stress can promote the denaturation of proteins. It was hypothesized that *HSP* can stabilize proteins that have been denatured owing to cold stress to refold the proteins so as to restore their functions. Yu et al. [80] cloned an *OsHsp* gene from rice, which was designated *Oshsp16.9* based on the molecular weight of the protein. A real-time PCR analysis showed that the expression of the *Oshsp16.9* gene was rapidly and strongly induced by stresses, including high salinity (250 mM NaCl), osmotic stress (300 mM mannitol), 100 μM ABA, cold (4 °C), and heat (45 °C); the overexpression of *Oshsp16.9* in rice conferred tolerance to salt in the transgenic plants, cold, and drought stress [80]. Although *HSPs* have been reported in many plants, most of them change plant stress resistance based on changes in overexpression. How the *HSPs* improve plant resistance through specific signal transduction mechanisms remains unclear. Currently, there are few reports on the role of *HSPs* in stress resistance in maize in China and throughout the world. The follow-up research to this study will focus on exploring the mechanism of action, protein evolution, and molecular relationship of *ZmHSPs* with the maize cold response, which will provide theoretical guidance for the molecular-assisted breeding of maize cold tolerance. It is worth noting that the *cis*-element analysis in the promoter region of the candidate gene showed that there was one *cis*-acting element that contained the *MYB* TF and bound the recognition sites, and seven *cis*-acting elements that were involved in the response to hormones. There were five *cis*-acting elements that defended against stress, three *cis*-acting elements that were related to the light response, and one *cis*-acting element that was related to the expression of the meristem (Appendix A). It is hypothesized that the candidate gene *Zm00001d012321* could be involved in mitigating cold stress in the MILs by regulating the hormone signal transduction system.

## 4. Materials and Methods

### 4.1. Plant Materials and Treatments

In this study, the MIL was synthesized using maize inbred lines B73, Mo17, Zheng58, and Chang7–2 as recurrent parents as the experimental materials. Healthy seeds (20–30 seeds/pot) were sown in flowerpots (21 cm × 16 cm) that contained a matrix that consisted of a mixture of peat, pine needles, and yellow clay. When the first leaf had fully expanded, the seedlings with inconsistent growth were pulled out of the pots with three replicates. At the single-leaf stage, all the plant materials were transferred into an artificial climate control chamber (MLR-352H-PC; Panasonic Healthcare Company of North America, Chicago, IL, USA) in the Laboratory of Cellular Genetics of the Maize Research Institute of Sichuan Agricultural University (Chengdu, China). The low- and control-temperature treatments were set at 2 and 25 °C, respectively, with 3000 μmol·m^−2^·s^−1^ illumination and 75% humidity. After the treatments had been applied for 3 and 5 d and plants had recovered for 24 h, the leaves of all the samples of MIL-IB030 and B73 were frozen in liquid nitrogen and stored at −80 °C for RNA-Seq with three biological repeats.

In addition, F_1_ was obtained by crossing cold-tolerant MIL-IB030 and cold-sensitive MIL-IB021, and the F_1_ continued to self-cross to obtain F_2_. An F_2_ ear with a larger number of grains was selected, and all the seeds were grown using single-grain sowing. A total of 289 F_2:3_ populations were constructed by self-crossing F_2_. A total of 30 seeds were selected from each F_2:3_ population, disinfected with 75% alcohol for 1 min, treated with 10% hydrogen peroxide for 10 min, washed with ddH_2_O three times, and finally placed neatly on pre-disinfected germinating paper (10 × 15; Anchor Company, St. Paul, MN, USA). The germinating papers were rolled up, fixed, numbered, and placed in an incubator (adjusted to 25 °C) that contained one-half Hoagland nutrient solution to initiate germination. Low-temperature (2 °C) stress was applied at the three-leaf and one-heart stages. There were three biological repeats for all the experiments.

### 4.2. Phenotypic Data Analysis

The phenotypes of all the materials tested were recorded after applying treatments from a minimum of 10 seedlings with three biological repeats. The PH, SFW, SDW, RFW, RDW, REC, RL, and RL/PH were measured. Fresh weight was the weight after absorbing the surface water of roots and stems, and the dry weight was the weight of roots and stems after drying at 100 °C for 2 h, followed by drying at 80 °C to reach a constant weight. Later, the membership function method was used to evaluate the cold tolerance of four sets of MILs using the following equations:U*_ij_* = (X*_ij_* − X*_jmin_*)/(X*_jmax_* − X*_jmin_*)(1)
U*_ij_* = 1 − (X*_ij_* − X*_jmin_*)/(X*_jmax_* − X*_jmin_*)(2)
where U*_ij_* represents the cold resistance membership value of the j index of i material; X*_ij_* represents the measured value of the j index of i material; X*_jmin_* represents the minimum value of all the j indices of the material, and X*_jmax_* represents the maximum value of all the j indices of the material. If the measured index was positively correlated with the cold resistance of a material, Equation (1) was used. Otherwise, Equation (2) was used. The specific membership values of all the indices accumulated to obtain an average value for comparison. If an average value was larger, it indicated that the inbred line was more resistant to cold.Type with high cold resistance (membership value >0.70);Type with moderate cold resistance (membership value between 0.60 and 0.70);Cold resistance type (membership value between 0.50 and 0.59);Highly sensitive type (membership values between 0.30 and 0.49);Highly cold-sensitive type (membership value <0.30).

Microsoft Excel (Redmond, WA, USA), SPSS (IBM, Inc., Armonk, NY, USA), and GraphPad Prism v. 9.0 (San Diego, CA, USA) were used for descriptive statistics, to test significance, and to obtain graphs of the analyzed data, respectively. The Performance Analytics package of a partial correlation analysis program in the R language was used to perform correlation, genetic rules, and PCA of the phenotypic traits.

### 4.3. Construction of the Genetic Map and QTL Mapping

Total DNA was extracted from fresh leaves using the CTAB method. A total of 2,396 SSR primers that were evenly distributed in 10 linkage groups were selected from the Maize GDB database (http://www.maizegdb.org, V4 (accessed on 1 March 2019)), and the polymorphism between parents MIL-IB030 and MIL-IB021 was screened. The SSR primers were synthesized by the Hangzhou Youkang Biotechnology Co., Ltd. (Hangzhou, China) and Beijing Qingke Biotechnology Co., Ltd. (Beijing, China). The genotypes of parent and F_2:3_ populations were determined by amplifying the DNA with SSR primer pairs and separating the PCR products on PAGE. The corresponding genotypes were used to calculate the polymorphism between parents and F_2:3_ populations. Briefly, the PAGE bands of SSR molecular markers were digitized. The same band type as the parental MIL-IB030 was designated 2; the same band type as the parental MIL-IB021 was designated 0; the same as the F_1_ band type was designated 1, and a missing band was designated −1. QTL ICI Mapping 4.2 software was used to construct a high-density genetic linkage map to cover the whole maize genome.

QTL ICI Mapping 4.2 software and CIM were used to locate the QTLs for the related traits in the 289 F_2:3_ populations. The LOD threshold was set at 3.0, and the primary and the epistatic interaction effects of the QTLs were analyzed.

### 4.4. RNA-Seq and Identification of the Candidate Gene

MIL-IB030 and recurrent parent B73 were treated at 2 °C and 25 °C for 3 d (72 h) and 5 d (120 h), followed by recovery at 25 °C for 24 h, with three biological repeats. Beijing Nuohezhiyuan Technology Co., Ltd. (Beijing, China) then extracted the RNA, constructed the cDNA library, and sequenced the genes (Illumina HiSeq^TM^ 4000; Illumina, San Diego, CA, USA) for this study. The resulting clean reads were compared with the maize reference genome (B73, V4 version) by TopHat version 2.0.10, and the differential gene expression and FPKM value of the target gene were calculated.

The WGCNA package in the R Studio software was used to process the FPKM of candidate genes from QTL mapping [81]. The parameters of the WGCNA program were as follows: variance data expression >0; no missing data expression <0.1; soft threshold = 10 (estimate value); max block size = 200; deep split = 4; min module size = 10; and merge cut height = 0.2. In each module, the genes with eigengene-based connectivity value (|KME|) > 0.9 and topological overlap measure (TOM) value > 0.2 were regarded as key candidate genes. Functional annotations of the candidate genes were obtained by searching the NCBI website (RefGen_v4) (https://www.ncbi.nlm.nih.gov/ (accessed on 1 March 2019)). In addition, BLAST [82] was used for the functional annotation of the candidate genes in the NT, COG, KOG, and Swiss-Prot databases. The GO seq method [83] was used for GO biological function enrichment analysis of differential gene sets, and the KOBAS method [84] was used for the KEGG metabolism and signal transduction pathway enrichment for differential genes. Both of these analyses used padj <0.05 as the threshold for significant enrichment.

### 4.5. Analyses of Gene Structure and Cis-Control Elements

The plant database (Ensembl Plants: http://plants.ensembl.org/ (accessed on 1 March 2019)) was used to search the candidate gene ID, and the structural annotation information of the gene was downloaded. A statistical analysis was then performed to determine the gene name, gene length, exons (start, stop), and introns based on the sequence information. To analyze the promoter of candidate genes, the upstream 2 kb sequence of each gene was downloaded, and the *cis*-acting regulatory elements were determined by submitting sequences to the Plant CARE database (http://bioinformatics.psb.ugeniant.Be/webtools/plantcare/html (accessed on 1 March 2021)). After sorting the data of the *cis*-acting regulatory elements that were obtained, the elements with functional annotation or more attention were retained. The core promoters with no actual function, including the CAAT-box and TATA-box, were eliminated. This study only retained the four categories of *cis*-regulatory elements that were related to hormones, the light response, environmental stress, and growth and development.

### 4.6. Cloning and Analysis of the Pattern of Expression of the Candidate Genes

The CDS sequences of the candidate genes (*Zm00001d037590* and *Zm00001d012321*) that were downloaded from the Maize GDB database (B73_V4) were used to design specific primer pairs using Primer 3.0 Amplify software (Appendix A). cDNA of the MIL-IB030 and B73 inbreds was used as templates to amplify the genes by PCR, and the PCR products were sequenced by the Beijing Qingke Biotechnology Co., Ltd. (Beijing, China). The BLAST function of the maize genome (B73_V4) was used to confirm the gene sequences, and SNP/Indel mutation sites in the CDS of candidate genes in the mutant and WT were then determined to observe whether these differences caused the changes in encoded amino acids.

Total RNA was extracted from the leaves of the seedlings of MIL-IB030 and B73 using a HiPure Plant RNA Mini Kit (Magen Biotech Co., Ltd., Guangzhou, China), according to the manufacturer’s instructions, after the plants had been treated at 25 °C and 2 °C for 0, 2, 6, 12, and 24 h. After confirming the concentration and quality of RNA using a NanoVue Plus nucleic acid protein analyzer (Eppendorf, BioPhotometer, China), the total RNA was reverse-transcribed to cDNA using a RevertAid First Strand cDNA Synthesis Kit (TaKaRa, Dalian, China), according to the manufacturer’s instructions. A CFX96 real-time PCR system (Bio-Rad, Hercules, CA, USA) was used to quantify gene expression with three biological replicates and four technical replicates (Appendix A). The real-time PCR data were analyzed using the comparative C_T_ method [85], and the accurate normalization of real-time quantitative RT-PCR data was revised by the geometric averaging of multiple internal control genes [86]. The *ZmACTIN* (*Zm00001d010159*) and *ZmGAPDH* (*Zm00001d049641*) genes were used as the internal control genes, and the primers used for qRT-PCR are listed in Appendix A. In this study, the MIQE value was determined as described by Vandesompele [87], and the detailed information is listed in Appendix A.

### 4.7. Subcellular Localization and Functional Verification of the Candidate Genes

PCAMBIA1302 was constructed using the restriction sites *Nco*I–*Spe*I for subcellular localization by Beijing Qingko Biotechnology Co., Ltd. *Zm00001d037590*-GFP was then transformed into Ben’s tobacco. GFP fluorescence was observed by confocal microscopy (ZEISS710; Carl Zeiss, Jena, Germany) with an excitation light value of 488.

EMS single-base mutants of *Zm00001d037590* and *Zm00001d012321* were purchased from Shandong Qilu Normal University (Shandong, China). Among them, EMS4-13c465 (G/A) was a single-base mutant of *Zm00001d037590*; EMS4-15d91a (C/T), EMS4-0b1833 (G/A), and EMS4-15d90b (G/A) were single-base mutants of *Zm00001d012321* [88]. The phenotypes of WT B73, the EMS single-base mutant, and MIL-IB030 were evaluated under low-temperature stress. In particular, the seedlings were treated at a low temperature when they had grown to the three-leaf and one-heart stages. The low-temperature treatment was established at 2 °C, and the control temperature was established at 25 °C. After 5 d of treatment, the REC of the leaves was measured, and the leaf RNA of the wild-type B73, MIL-IB030, and EMS mutant were extracted for validation by qRT-PCR.

### 4.8. Identification of the Gene Family among Different Maize Varieties

In this study, the homologous gene families of *Zm00001d012321* and *Zm00001d037590* in 15 different maize inbred lines were identified and analyzed phylogenetically and for their selection pressure. Among them, all the data of different inbred lines of maize, including B97, CML52, CML247, DK105, Ki3, Ki11, Ky21, Mo17, Mo18, Ms71, NC350, Oh43, P39, W22, and Mexican feathergrass (*Nassella tenuissima*) (*Zx-PI566673*), were downloaded from the Maize GDB database.

The homologous genes of *Zm00001d012321* and *Zm00001d037590* were identified by the similarity between the target and aligned sequences, and the genes whose percentage identity between the aligned sequence and the target sequence >60% were classified as paralogous genes. BLAST 2.9.0 was used to construct a BLAST database using the protein sequences that corresponded to the longest transcripts extracted from the genomes of inbred lines of each maize species, and they were then compared with the protein sequences of the longest transcripts of *Zm00001d012321* and *Zm00001d037590*. The parameters of comparison were as follows: score value ≥ 100, E < 1e^−5^, and an output format of ‘6′ to identify the *Zm00001d012321* and *Zm00001d037590* homologous gene sequences. MEGA_X_10.0.5_win 64 was used to construct a phylogenetic tree. First, the protein sequences of *Zm00001d012321* and *Zm00001d037590* were aligned with ClustalW, and the alignment parameters were set to default parameters. After the comparison results were obtained, the ML method based on the Poisson correction model was used to construct the phylogenetic tree. Bootstrap was used for detection, and the parameter was set to 1000 times. The CDS of the gene family members were aligned. The sequence alignment was performed using the ClustalW method of MEGA X, and the selection pressure analysis was performed on the alignment results using the KaKs_Calculator 2.0. YN was selected as the model parameter. The genes were divided into three categories based on the size of Ka/Ks and counted. The division principles were as follows: Ka/Ks > 1, positive selection; Ka/Ks = 1, neutral evolution; Ka/Ks < 1, purify selection.

## Figures and Tables

**Figure 1 ijms-24-02629-f001:**
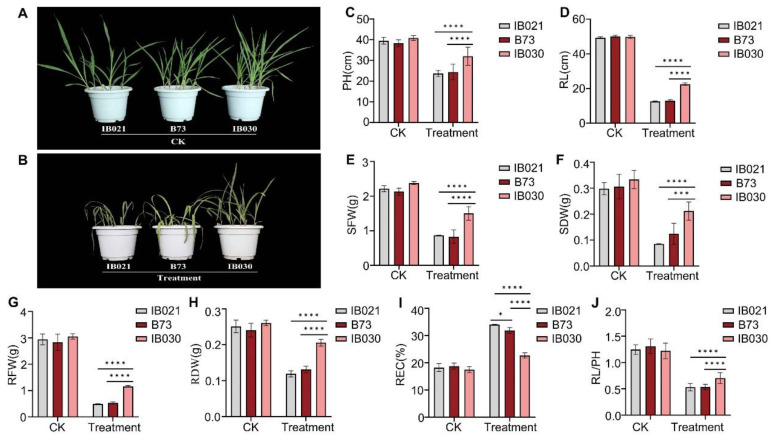
Phenotypes and statistical comparison of cold-resistant MIL-IB030, cold-sensitive MIL-IB021, and recurrent parent B73 under normal and low-temperature stress conditions. Note: (**A**) and (**B**) display plant phenotypes. (**C**)–(**J**) show statistical comparison of plant’s performance-related indicators, including plant height (PH), seedling fresh weight (SFW), seedling dry weight (SDW), root fresh weight (RFW), root dry weight (RDW), relative electrical conductivity (REC), root length (RL), and RL/PH, grown at 25 °C for 5 d and 2 °C for 5 d. * indicates significance at 0.05 level, *** indicates significance at 0.001 level, and **** indicates significance at 0.0001 level.

**Figure 2 ijms-24-02629-f002:**
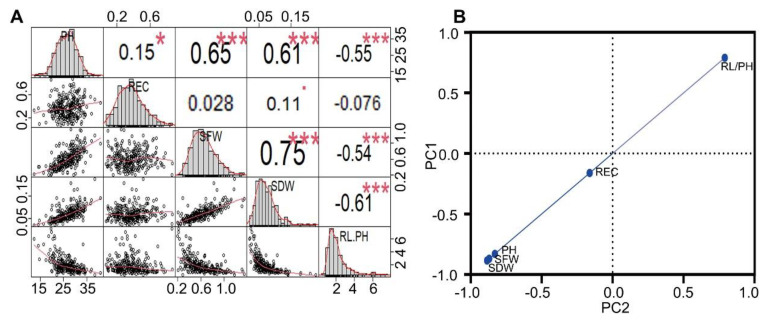
(**A**) A genetic analysis of phenotypic indexes of 289 F_2:3_ populations under cold stress. (**B**) The phenotypic indices of 289 F_2:3_ populations under low-temperature stress were analyzed by principal component analysis. Note: PH, SFW, SDW, REC, and RL represent plant height, seedling fresh weight, seedling dry weight, relative electrical conductivity, and root length, respectively. * indicates significance at 0.05 level, and *** indicates significance at 0.001 level.

**Figure 3 ijms-24-02629-f003:**
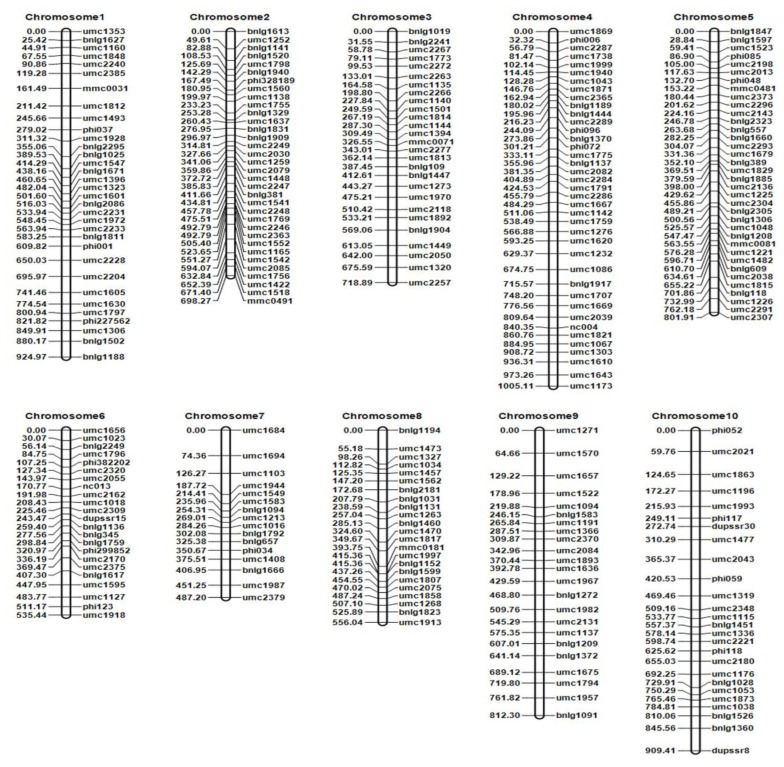
A genetic map of phenotypic indicators of the F_2:3_ population. The left side of each chromosome in the figure represents the genetic distance between markers in centimorgans (cM), and the right side shows the SSR markers’ names on the respective chromosome. Note: LOD score represents the LOD value of each QTL, while additive effect represents the additive effect value.

**Figure 4 ijms-24-02629-f004:**
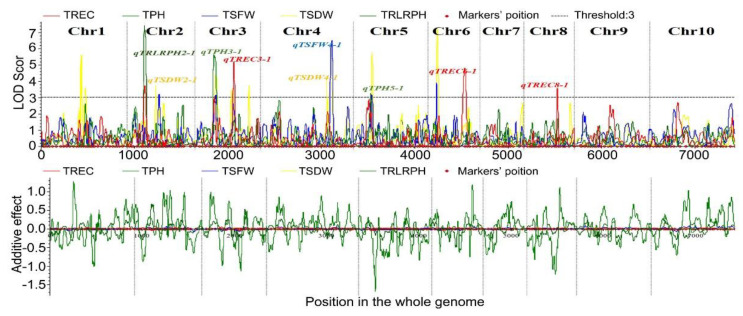
QTL mapping results of seedling-related traits in F_2:3_ populations (2 °C, 5 d). Note: PH, SFW, SDW, REC, and RL represent plant height, seedling fresh weight, seedling dry weight, relative electrical conductivity, and root length, respectively.

**Figure 5 ijms-24-02629-f005:**
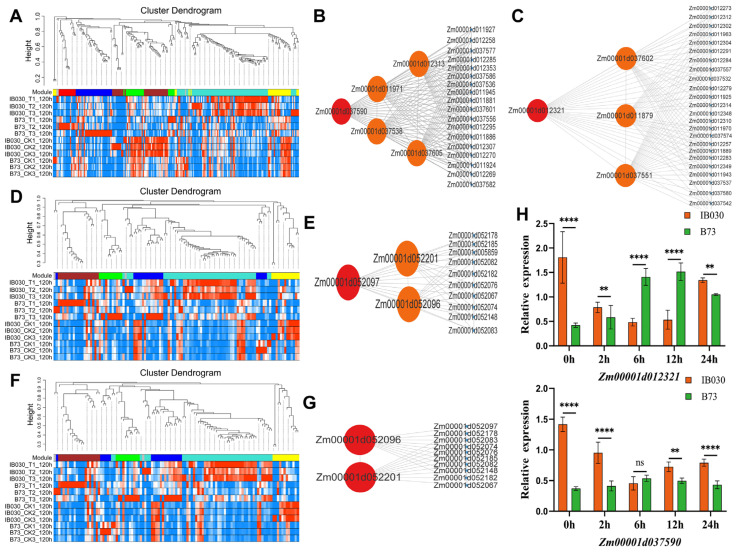
The identified modules and regulatory networks of hub genes in this study. Note: (**A**) Hierarchical clustering tree diagram of modules; (**B**) green module showing the regulatory network of the 5 hub genes; (**C**) regulatory network of the 4 hub genes in the yellow module; (**D**) hierarchical clustering tree diagram of modules; (**E**) regulatory network of a hub gene in the yellow module; (**F**) hierarchical clustering tree diagram of modules; (**G**) regulatory network of two hub genes in the yellow module; (**H**) relative expression analysis of candidate genes. Note: SFW, SDW, and REC represent seedling fresh weight, seedling dry weight, and relative electrical conductivity, respectively. ** indicates significance at 0.01 level, **** indicates significance at 0.001 level, and ns indicates not significance.

**Figure 6 ijms-24-02629-f006:**
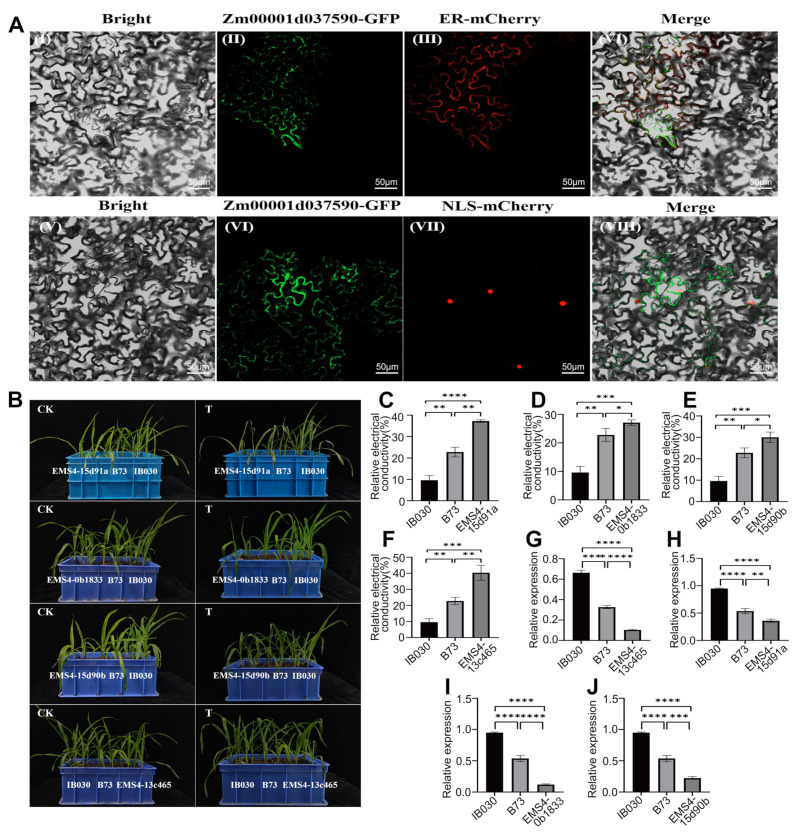
(**A**) Subcellular localization of *Zm00001d037590* protein. I and V were bright fields; II and VI were *Zm00001d037590*-GFP; III and VII were ER-mCherry marker and NLS-mCherry marker, respectively; IV was the green fluorescence of Zm00001d037590-GFP fusion protein, which coincides with the red fluorescence of ER-mCherry; VIII was green fluorescence of Zm00001d037590-GFP fusion protein, which did not coincide with the red fluorescence of NLS-mCherry. (**B**) Growth status of wild-type B73, EMS single-base mutants, and MIL-IB030 before and after low-temperature stress. (**C**)–(**F**) REC of the leaves of wild-type B73, EMS single-base mutant, and MIL-IB030 after cold stress. (**G**)–(**J**) The relative expression of key candidate genes in wild-type B73, EMS single-base mutant, and MIL-IB030 under low-temperature stress. Note: EMS4-13c465 (*Zm00001d037590*), EMS4-15d91a (*Zm00001d012321*), EMS4-0b1833 (*Zm00001d012321*), and EMS4-15d90b (*Zm00001d012321*). * indicates significance at 0.05 level, ** indicates significance at 0.01 level, *** indicates significance at 0.001 level, and **** indicates significance at 0.0001 level.

**Figure 7 ijms-24-02629-f007:**
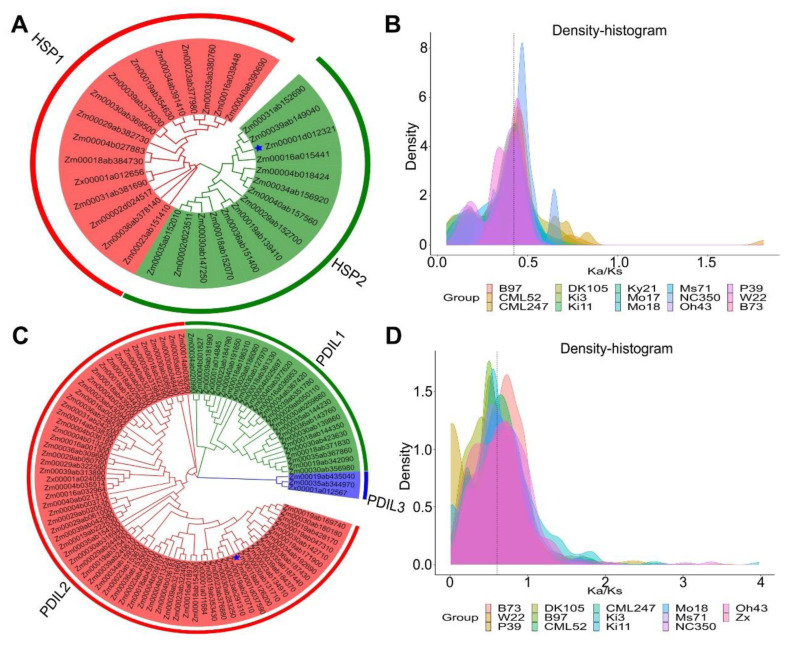
(**A**) Phylogenetic tree of homologous genes in *Zm00001d012321*. (**B**) Analysis of homologous gene selection pressure in *Zm00001d012321*. (**C**) Phylogenetic tree of homologous genes in *Zm00001d037590*. (**D**) Analysis of homologous gene selection pressure in *Zm00001d037590*.

**Table 1 ijms-24-02629-t001:** Statistical analysis of phenotypic indicators of F_2:3_ population (2 °C; 5 d).

Trait	Range	Mean	SE	CV/%	Kurtosis	Skewness	Sig
PH(cm)	12.40–40.93	26.9065	0.2546	0.0095	0.3284	−0.1786	**
REC	0.06–0.87	0.3726	0.0096	0.0256	−0.0803	0.5014	**
SFW(g)	0.20–1.35	0.6162	0.0130	0.0211	0.2186	0.5765	**
SDW(g)	0.02–0.23	0.0733	0.0018	0.0240	3.9244	1.3964	**
RL/PH	0.40–7.63	1.8908	0.0682	0.0361	5.6565	2.1093	**

Note: ** indicates significance at 0.01 level. Plant height (PH), seedling fresh weight (SFW), seedling dry weight (SDW), relative electrical conductivity (REC), and root length (RL).

**Table 2 ijms-24-02629-t002:** Summary of main QTLs controlling seedling traits of F_2:3_ population (2 °C; 5 d).

Trait	QTL	Physical Distance/Mb	Chromosome	Marker Interval	LOD	PVE/%	Additive Effect
REC	*qTREC3-1*	2.10	3	umc1892–umc2118	3.08	2.09	0.0383
REC	*qTREC6-1*	3.41	6	bnlg2249–umc1918	4.82	3.46	0.0035
REC	*qTREC8-1*	9.09	8	umc1268–bnlg1152	3.57	6.03	0.0046
PH	*qTPH3-1*	–	3	umc1892–umc2118	5.65	8.48	−0.5848
PH	*qTPH5-1*	–	5	umc2296–umc2143	3.25	6.62	−1.6670
SFW	*qTSFW4-1*	5.52	4	umc1620–umc1667	6.51	8.77	0.0304
SDW	*qTSDW2-1*	0.06	2	bnlg1329–umc1637	3.72	1.00	0.0055
SDW	*qTSDW4-1*	5.52	4	umc1620–umc1667	3.99	1.41	0.0076
RL/PH	*qTRLRPH2-1*	1.25	2	umc1798–bnlg1940	7.41	10.57	0.6498

Note: PH, SFW, SDW, REC, and RL represent plant height, seedling fresh weight, seedling dry weight, relative electrical conductivity, and root length, respectively.

**Table 3 ijms-24-02629-t003:** Details of hub genes detected by WGCNA.

Trait	Module	Gene	KME	TOM	Gene Description
SDW	yellow	*Zm00001d052201*	0.96	0.25	probable magnesium transporter NIPA8
SDW	yellow	*Zm00001d052096*	0.94	0.27	uncharacterized LOC103654201
SDW	yellow	*Zm00001d052097*	0.99	0.25	uncharacterized LOC103654201
REC	green	*Zm00001d012313*	0.9	0.39	uncharacterized LOC100285836
REC	green	*Zm00001d037538*	−0.98	0.33	arginyl-tRNA synthetase
REC	green	*Zm00001d037590*	0.91	0.29	protein disulfide isomerase 7
REC	green	*Zm00001d037605*	0.92	0.26	GATA transcription factor 23
REC	green	*Zm00001d011971*	0.96	0.21	delta3,5-delta2,4-dienoyl-CoA isomerase
REC	yellow	*Zm00001d012321*	−0.97	0.43	nematode resistance protein-like HSPRO1
REC	yellow	*Zm00001d037551*	0.96	0.35	putative GTP diphosphokinase CRSH chloroplastic
REC	yellow	*Zm00001d011879*	−0.95	0.31	uncharacterized LOC100275154
REC	yellow	*Zm00001d037602*	0.92	0.27	U1 small nuclear ribonucleoprotein A
SFW	yellow	*Zm00001d052201*	0.97	0.52	probable magnesium transporter NIPA8
SFW	yellow	*Zm00001d052096*	0.94	0.51	uncharacterized LOC103654201

## Data Availability

The datasets presented in this study can be found in online repositories and the Appendix A. The transcriptome data can be found in the National Genomics Data Center (NGDC) database. Readers can query the transcriptome data by visiting the link (https://ngdc.cncb.ac.cn/search/?dbId=gsa&q=CRA007319, accessed on 23 March 2021) (BioProject: PRJCA010234; Accession number: CRA007319).

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
