# Peer review of "QTL Mapping and a Transcriptome Integrative Analysis Uncover the Candidate Genes That Control the Cold Tolerance of Maize Introgression Lines at the Seedling Stage"

_ijms, 2023, doi:10.3390/ijms24032629_

Round 1
Reviewer 1 Report
MDPI ijms-2065263
Identification of Candidate Genes That Control the Chilling Tolerance of Maize Introgression Lines at the Seedling Stage Using QTL-Mapping and Transcriptomics Integrative Analysis
The manuscript presents crucial failures and gaps. To be published in this journal, the manuscript must significantly increase its quality.
Although the introduction is well written, it needs substantial improvements supported with current literature. Likewise, the discussion is not sufficiently supported with current and appropriate references.
An exhaustive analysis of lipids already demonstrated to be cold-responsive is compulsory. For details please revise: doi: 10.1093/pcp/pcaa139; doi: 10.1093/pcp/pcn173; and doi: 10.1093/pcp/pcp051, among others.
Regarding the gene expression analyses, authors should have applied the appropriate protocols to analyzing real-time PCR data by the comparative C(T) method (doi:10.1006/meth.2001.1262 and doi: 10.1038/nprot.2008.73) and revised the accurate normalization of real-time quantitative RT-PCR data by geometric averaging of multiple internal control genes (doi: 10.1186/gb-2002-3-7). Authors do not exactly explain how the determined the reference genes for the whole analysis, and do not discuss the determination of the M value according to the protocol described by Vandesompele et al. 2002: https://doi.org/10.1186/gb-2002-3-7-research0034
All figures need substantial improvements. They can be hardly seen because of the size they are currently presented in the current version of the manuscript.
In general, the manuscript may be substantially improved from English review and editing.
If authors are willing to answer all comments and concerns made, they can resubmit a revised version of the manuscript as soon as possible, addressing all failures and gaps detected during the review process.
Author Response
List of Responses
Thank you for your letter, and for your careful review and constructive suggestions with regard to manuscript (ID: ijms-2065263). Those comments are valuable and very helpful for us to revise and improve our paper. We have studied these comments carefully and have substantially revised our manuscript. Revised portions are marked in red in the paper. Our main corrections in the paper and the responses to the comments are as follows.
Reviewer 1:
(1) The manuscript presents crucial failures and gaps. To be published in this journal, the manuscript must significantly increase its quality.
Response: Thank you very much for reviewer's valuable comments. We have made detailed modify to the content of the manuscript, such as introduction, figure, discussion and English.
(2) Although the introduction is well written, it needs substantial improvements supported with current literature. Likewise, the discussion is not sufficiently supported with current and appropriate references.
Response: First of all, I would like to thank the reviewer for your praise on the writing of the introduction. Secondly, thank you very much for reviewer's valuable comments. We have supplemented the literature in the introduction by adding some current published articles, and the details are as follows: P59-P75. Finally, we add the recent literature to the discussion section, and the details are as follows: P534-P553, P583-P601, P616-P629, and P636-P647.
(3) An exhaustive analysis of lipids already demonstrated to be cold-responsive is compulsory. For details please revise: doi: 10.1093/pcp/pcaa139; doi: 10.1093/pcp/pcn173; and doi: 10.1093/pcp/pcp051, among others.
Response: Thank you for pointing this out, and we are sorry that we did not describe this part clearly. We have supplemented this part in detail in the manuscript, and the details are as follows: P77-P81, P92-P96, and P101-P113.
(4) Regarding the gene expression analyses, authors should have applied the appropriate protocols to analyzing real-time PCR data by the comparative C(T) method (doi:10.1006/meth.2001.1262 and doi: 10.1038/nprot.2008.73) and revised the accurate normalization of real-time quantitative RT-PCR data by geometric averaging of multiple internal control genes (doi: 10.1186/gb-2002-3-7). Authors do not exactly explain how the determined the reference genes for the whole analysis, and do not discuss the determination of the M value according to the protocol described by Vandesompele et al. 2002: https://doi.org/10.1186/gb-2002-3-7-research0034.
Response: Thank you very much for reviewer's valuable comments. Regarding the gene expression analyses and how the determined the reference genes for the whole analysis: First of all, we have made modifications in the part of materials and methods, and the details are as follows: P780-P797. Secondly, we also modified the analysis part of the results, and the details are as follows: Figure 3. Finally, the M value is described in detail according to the protocol description, and the details are as follows: Supplementary 6.
(5) All figures need substantial improvements. They can be hardly seen because of the size they are currently presented in the current version of the manuscript.
Response: Thank you very much for reviewer's valuable comments. We have modified all the figure of the manuscript and uploaded them to the system in the form of zip.
(6) In general, the manuscript may be substantially improved from English review and editing.
Response: First of all, I would like to thank the reviewers for their valuable comments. Secondly, we have revised and edited the manuscript in English.
Supplementary Paper 6 the determination of the M value according to the protocol described
- Experimental design
1) Definition of experimental and control groups
Experimental groups: MIL-IB030 and B73 were treated at 2℃ for 0h, 2h, 6h, 12h and 24h at seedling stage, respectively
Control groups: MIL-IB030 and B73 were treated at 25℃ for 0h, 2h, 6h, 12h and 24h at seedling stage, respectively
2) Number within each group
Experimental groups: 15 samples
Control groups: 15 samples
- Sample
1) Description
RNA from the leaves of MIL-IB030 and B73 at seedling stage were treated at 25℃ and 2℃ for 0h, 2h, 6h, 12h and 24h, respectively.
2) Micro-dissection or macro-dissection
Macro-dissection
3) Processing procedure
The leaves of MIL-IB030 and B73 were collected after treatment at 25℃ and 2℃ for 0h, 2h, 6h, 12h and 24h, respectively.
4) If frozen, how and how quickly?
The samples were frozen in liquid nitrogen for 15 min
5) If fixed, with what and how quickly?
No
6) Sample storage conditions and duration
The samples were stored in -80 ℃ ultra-low temperature refrigerator for 1h.
- Nucleic acid extraction
1) Procedure and/or instrumentation
Total RNA was extracted from leaves of the seedling of MIL-IB030 and B73 after treating them at 25 °C and 2 °C for 0 h, 2 h, 6 h, 12 h, and 24 h using HiPure Plant RNA Mini Kit (Magen Biotech Co., Ltd) following manual instruction. After confirming the concentration and quality RNA by NanoVue Plus nucleic acid protein analyzer (Eppendorf, BioPhotometer, China), the total RNA was reverse transcribed to cDNA by using RevertAid First Strand cDNA Synthesis Kit (TaKaRa, Dalian, China) following manual instruction.
2) Name of kit and details of any modifications and Details of DNase or RNase treatment
Table 1 Remove genomic contamination system in RNA
|
Remove genomic contamination system in RNA |
|
|
Reagent |
Volume (μL) |
|
10X Reaction Buffer |
1 |
|
with MgCl2 |
1 |
|
total RNA |
1 |
|
DNase I |
1 |
|
Water, nuclease-free |
7 |
|
Total |
10 |
3) Contamination assessment (DNA or RNA) and Method/instrument
RNA concentration and OD260 /OD280 values were detected by NanoVue Plus nucleic acid protein analyzer, and RNA sample quality was tested.
- Reverse transcription
1) Complete reaction conditions
SynthesisKit cDNA synthesiskit (TaKaRa) was used RevertAid First Strand cDNA SynthesisKit kit (Takara). Total RNA of samples was used as template for reverse-transcription into cDNA. Genomic contamination was removed from RNA at the same time. Add 1 µL of 50 mM EDTA and incubate at 65℃ for 10 min
2) Amount of RNA and reaction volume
Table 2 Reverse transcription system
|
Reverse transcription system |
|
|
Reagent |
Volume (μL) |
|
5X Reaction Buffer |
4 |
|
Oligo (dT) 18 primer |
1 |
|
10 mM dNTP Mix |
2 |
|
total RNA |
1 |
|
RiboLock Rnase Inhibitor |
1 |
|
RevertAid M-MuLV RT |
1 |
|
Water, nuclease-free |
10 |
|
Total |
20 |
3) Priming oligonucleotide (if using GSP) and concentration
No
4) Reverse transcriptase and concentration
RiboLock Rnase Inhibitor
5) Temperature and time
The reverse transcription procedure was performed at 42℃ for 60 min. After 5 min at 72℃ and 12℃, the samples were stored at -20℃ for use.
- qPCR target information
1) Gene symbol
Zm00001d012321 and Zm00001d037590
2) Sequence accession number
Zm00001d012321: LOC100382754
Zm00001d037590: LOC606414
3) Amplicon length
Zm00001d012321: 1425bp
Zm00001d037590: 4643bp
4) Location of each primer by exon or intron (if applicable)
Zm00001d012321: The first exon
Zm00001d037590: The first exon
- qPCR oligonucleotides
1) Primer sequences
|
Gene |
Forward primer |
Reverse primer |
|
Zm00001d012321 |
TTATCGTTCTCGAGGCCAGC |
GGCTCCGGTTGAACTCTCTC |
|
Zm00001d037590 |
TTCTGCAGCCAAGCATGGAA |
CCGTAAAGTGGTGGTTCGCT |
2)RT Primer DB identification number
No
- qPCR protocol
1) Complete reaction conditions, Reaction volume and amount of cDNA/DNA, and Primer, probe, Mg2+, and dNTP concentrations
Table 3 RT-PCR system
|
Reagent |
Volume (μL) |
|
ddH2O2 |
5 |
|
2X SYBR Green Fast qPCR Mix |
7.5 |
|
Forward primer(10μM) |
0.4 |
|
Reverse primer(10μM) |
0.4 |
|
cDNA |
1 |
|
Total |
10 |
4) Polymerase identity and concentration
2X SYBR Green Fast qPCR Mix
5) Buffer/kit identity and manufacturer
No
6) Complete thermocycling parameters
Table 4 RT-PCR reaction procedure
|
Step |
Temperature |
Time |
|
1 |
95 ℃ |
3 min |
|
2 |
95 ℃ |
5s |
|
3 |
60 ℃ |
30 s |
|
4 |
﹢H10Plate Read,Go to step 2 |
40-45 cycles |
|
5 |
Melt Curve 65-95℃ |
|
|
6 |
﹢Plate Read,increment 0.5℃ |
0.5 s |
|
7 |
end |
  |
7)Manufacturer of qPCR instrument
The CFX96 real-time PCR system (Bio-Rad)
- Data analysis
1) qPCR analysis program (source, version)
The real-time PCR data were analyzed by the comparative CT method (Schmittgen et al., 2008. doi:10.1038/nprot.2008.73).
2) Method of Cq determination
The Cq value is the number of PCR cycles at the intersection of the sample response curve and the threshold line. The PCR instrument software calculates the Cq value of each sample.
3) Justification of number and choice of reference genes
The ZmACTIN (Zm00001d010159) and ZmGAPDH (Zm00001d049641) genes were used as the internal control genes
4) Description of normalization method
Revised the accurate normalization of real-time quantitative RT-PCR data by geometric averaging of multiple internal control genes.
5) Number and stage (reverse transcription or qPCR) of technical replicates
Three biological replicates, four technical replicates.
6) Repeatability (intraassay variation)
There were four replicates within the group, and there was little variation.
7)Statistical methods for results significance
The real-time PCR data were analyzed by the comparative CT method, and revised the accurate normalization of real-time quantitative RT-PCR data by geometric averaging of multiple internal control genes.
8)Software (source, version)
Microsoft Excel (Redmond, WA, USA), SPSS (IBM, Inc., Armonk, NY, USA), and GraphPad Prism v9.0 (San Diego, CA, USA) were used for descriptive statistics.

Reviewer 2 Report
Review Report
Ø Make Crisp Heading of your Manuscript
Ø General impact of chilling injury needed to be quantified, magnitude of damage worldwide or native
Ø Abbreviations needs to be amply described
Ø The manuscripts requires proper formatting, by putting up things in right order etc
Author Response
List of Responses
Thank you for your letter, and for your careful review and constructive suggestions with regard to manuscript (ID: ijms-2065263). Those comments are valuable and very helpful for us to revise and improve our paper. We have studied these comments carefully and have substantially revised our manuscript. Revised portions are marked in red in the paper. Our main corrections in the paper and the responses to the comments are as follows.
Reviewer 2:
(1) Make Crisp Heading of your Manuscript
Response: Thank you very much for reviewer's valuable comments. We have changed the Heading of the manuscript, and the details are as follows: QTL-mapping and Transcriptome integrative analysis uncover the candidate genes controlling cold tolerance of maize introgression lines at the seedlings
(2) General impact of chilling injury needed to be quantified, magnitude of damage worldwide or native
Response: Thank you for pointing this out, and we are sorry that we did not describe this part clearly. We have supplemented this part in detail in the manuscript, and the details are as follows: P50-P55.
(3) Abbreviations needs to be amply described
Response: Thank you very much for reviewer's valuable comments. In the manuscript, we have added relevant content, and the details are as follows: P46.
(4) The manuscripts requires proper formatting, by putting up things in right order etc
Response: Thank you very much for the reviewer’s comments, and we have adjusted the format of the manuscript according to IJMS publishing standards.

Reviewer 3 Report
Comments to the authors
The manuscript clearly have value for publication. The data reported are highly informative. In order for the title to be properly scientific, I suggest that the authors could improve the title and add the “heavy” information, like transcriptome for example.
Author Response
List of Responses
Thank you for your letter, and for your careful review and constructive suggestions with regard to manuscript (ID: ijms-2065263). Those comments are valuable and very helpful for us to revise and improve our paper. We have studied these comments carefully and have substantially revised our manuscript. Revised portions are marked in red in the paper. Our main corrections in the paper and the responses to the comments are as follows.
Reviewer 3:
The manuscript clearly have value for publication. The data reported are highly informative. In order for the title to be properly scientific, I suggest that the authors could improve the title and add the “heavy” information, like transcriptome for example.
Response: First of all, thank you very much for the reviewer's praise and approval of the content of the manuscript. Secondly, thank you very much for reviewer's valuable comments. We have changed the Heading of the manuscript, and the details are as follows: QTL-mapping and Transcriptome integrative analysis uncover the candidate genes controlling cold tolerance of maize introgression lines at the seedlings.

Round 2
Reviewer 1 Report
MDPI ijms-2065263
Identification of Candidate Genes That Control the Chilling Tolerance of Maize Introgression Lines at the Seedling Stage Using QTL-Mapping and Transcriptomics Integrative Analysis
Authors have substantially improved the new revised version of the manuscript. However, critical issues still remain without being addressed yet, which were mentioned in the previous review corried out:
All figures need substantial improvements. They can be hardly seen because of the size they are currently presented.
In general, the manuscript may be substantially improved from English review and editing.
Author Response
List of Responses
Thank you for your letter, and for your careful review and constructive suggestions with regard to manuscript (ID: ijms-2065263). Those comments are valuable and very helpful for us to revise and improve our paper. We have studied these comments carefully and have substantially revised our manuscript. Revised portions are marked in red in the paper. Our main corrections in the paper and the responses to the comments are as follows.
Reviewer 1:
(1) All figures need substantial improvements. They can be hardly seen because of the size they are currently presented.
Response: Thank you very much for reviewer's valuable comments. All Figures (both Figures in manuscript and supplementary Figures) have been adjusted to the maximum to ensure that the Figures are clearly scaled, and the text is clear. In addition, we uploaded all the Figures separately into the system.
(2) In general, the manuscript may be substantially improved from English review and editing.
Response: Thank you very much for reviewer's valuable comments. As for the English improvement of this manuscript, the author invited a professional institution to revise the English, and we uploaded the proof of modification into the system.
